# Transcription Factor NFE2L1 Decreases in Glomerulonephropathies after Podocyte Damage

**DOI:** 10.3390/cells12172165

**Published:** 2023-08-29

**Authors:** Mustafa Elshani, In Hwa Um, Steve Leung, Paul A. Reynolds, Alex Chapman, Mary Kudsy, David J. Harrison

**Affiliations:** 1School of Medicine, University of St Andrews, St Andrews KY16 9TF, UKdavid.harrison@st-andrews.ac.uk (D.J.H.); 2Pathology, Laboratory Medicine, Royal Infirmary of Edinburgh, Little France, Edinburgh EH16 6NA, UK; 3NuCana plc, 3 Lochside Way, Edinburgh EH12 9DT, UK; 4Urology Department, Western General Hospital, Edinburgh EH4 2XU, UK; 5Urology Department, Victoria Hospital, Hayfield Road, Kirkcaldy KY2 5AH, UK

**Keywords:** podocytes, NFE2L1, NQO1, glomerular disease, transcription factor

## Abstract

Podocyte cellular injury and detachment from glomerular capillaries constitute a critical factor contributing to kidney disease. Notably, transcription factors are instrumental in maintaining podocyte differentiation and homeostasis. This study explores the hitherto uninvestigated expression of Nuclear Factor Erythroid 2-related Factor 1 (NFE2L1) in podocytes. We evaluated the podocyte expression of NFE2L1, Nuclear Factor Erythroid 2-related Factor 2 (NFE2L2), and NAD(P)H:quinone Oxidoreductase (NQO1) in 127 human glomerular disease biopsies using multiplexed immunofluorescence and image analysis. We found that both NFE2L1 and NQO1 expressions were significantly diminished across all observed renal diseases. Furthermore, we exposed human immortalized podocytes and ex vivo kidney slices to Puromycin Aminonucleoside (PAN) and characterized the NFE2L1 protein isoform expression. PAN treatment led to a reduction in the nuclear expression of NFE2L1 in ex vivo kidney slices and podocytes.

## 1. Introduction

Podocytes are terminally differentiated cells situated in the Bowman’s space within the glomerular tuft, wrapping around the glomerular capillaries with specialised interdigitating foot processes covering the entirety of the glomerular basement membrane (GBM) surface area. Podocytes form complex cellular processes around the capillaries creating the final filtration barrier; therefore, any injury or detachment has a pivotal role in the pathogenesis of the glomerular disease [1].

Podocytes sit within a dynamic environment in the glomerular tuft, constantly being subjected to changing conditions and cues on all sides. While maintaining a tight grip on the capillaries, they are also responsible for maintaining the GBM, in addition to adapting to shear fluidic stress on both the apical and basal sides of the cell. Because podocytes are terminally differentiated and cannot be replaced, they require effective mechanisms to protect the cells against excessive exposure to micro-molecules such as glucose that are filtered into the Bowman’s space. Different stages of the evolution of podocyte injury have been described [2]. Podocytes initially undergo foot process effacement defined by loss of the slit diaphragm. Since podocyte detachment is detrimental and cannot be readily repaired, effacement can be considered a self-preserving mechanism at the expense of its crucial function, that of a filtration barrier. Podocytes, however, have the ability to recover from injury, as has been demonstrated in vitro, where after removal of puromycin aminonucleoside (PAN), podocytes recovered slit diaphragm proteins and their biomechanical properties [3]. Further evidence of reversible injury is that 80% of patients with minimal change nephropathy (MCD), in which foot process effacement is the characteristic morphological feature, recover following corticosteroid treatment [4]. Even though podocytes recover from injury, they cannot sustain persistent insult, as this causes a gradual reduction in expression of their focal adhesion and slit diaphragm proteins, eventually leading to their detachment [5]. Podocyte loss occurs in diabetic nephropathy [6], treatment with chemotherapeutic agents [7], hypertension and old age [8].

Nuclear factor erythroid-derived 2-like 1 (NFE2L1), also referred to as NRF1, is a member of the CNC-bZIP family of transcription factors responsible for the transcription of antioxidant proteins via the antioxidant response element (ARE). Functionally, NFE2L1 and NFE2L2 (NRF2) proteins have distinct roles. Global knockout Nrf2 mice are viable, but have higher susceptibility to cancer than wild-type mice [9]. By contrast, the knockout of the Nfe2l1 gene causes oxidative stress and embryonic lethality [10]. Consequently, it has been concluded that, in mice, Nrf2 is mainly responsible for regulating the inducible expression of the antioxidant genes. At the same time, NFE2L1 is thought to be responsible for the basal expression of these genes [11]. In a basal state, the NFE2L1 isoform TCF11, with a mass of 140-kDa, is anchored at the endoplasmic reticulum (ER) membrane and undergoes rapid degradation via HRD1-mediated ubiquitination [12]. While docked, in the presence of glucose, it is glycosylated and yields an inactive glycoprotein. Under as-yet-unknown biological cues, NFE2L1 undergoes deglycosylation [13], retro-translocation [14] from the ER lumen and is proteolytically cleaved [15], which generates an active 95-kDa nuclear isoform heterodimerised with small musculoaponeurotic fibrosarcoma oncogene (sMAF) [16]. NAD(P)H:quinone oxidoreductase 1 (NQO1) is a cytosolic homodimeric flavoprotein, its main function being the detoxification of endogenous and exogenous quinones by two-electron reduction [17]. It is expressed highly in basal state podocytes in human renal biopsies [18] and is a transcriptional target of both NFE2L1 and NFE2L2 [19]. However, little is known about its function and regulation in podocytes. 

NFE2L1 expression and its post-translation processing are tightly regulated, which is crucial for its specific function. NFE2L1 mRNA has been identified in podocytes [20]; however, its protein expression has not been previously characterised; therefore, we investigated NFE2L1 protein expression in podocytes from kidney biopsies in different kidney diseases, and in addition we characterised the protein isoforms in an immortalised podocyte cell line. We found that the expression of NFE2L1 was drastically reduced in podocytes in biopsies from kidney disease patients. Additionally, podocytes exposed to PAN showed a decrease in the nuclear isoform of NFE2L1 in vitro and ex vivo.

## 2. Materials and Methods

### 2.1. Biopsies from Patients with Renal Disease

The disease cohort was selected from archived tissue biopsies obtained from the Royal Infirmary of Edinburgh from 2015 to 2018 (n = 127). Ethical approval was granted by NHS Lothian NRS Bioresource (15/ES/0094) and by the University of St Andrews teaching and research ethics committee (MD9202). Where feasible, tissue blocks were linked to clinical data at time of biopsy, and then fully de-identified. Cases included primary and secondary glomerular disease: Minimal Change Disease (MCD) n = 13, focal segmental glomerulosclerosis (FSGS) n = 37, diabetes mellitus n = 27, mesangiocapillary (membranoproliferative) glomerulonephritis (MCGN) n = 12, mesangial IgA/IgG nephropathy n = 37. As a control group, the cohort included 11 needle biopsies reported as morphologically normal or samples from nephrectomy of renal carcinoma (RCC) patients; samples were taken as far as possible from the tumour. Each case was reviewed by an experienced renal pathologist (D.J.H.). Details of the cohort are summarized in Table 1.

### 2.2. Ex Vivo Tissue Slice Preparation

In this study, six patients undergoing complete nephrectomy for RCC were recruited to participate. None of the recruited patients had a previous history of kidney disease. From each patient, a wedge of tissue sample approximately 1 cm^3^ in size was dissected from the normal kidney parenchyma for further analysis. The sample of kidney tissue was immediately placed into the M199, cat. #M4530 (Sigma-Aldrich, Paisley, UK) culture medium containing 5% Penicillin/Streptomycin, cat. #10378016 (Gibco™, Paisley, UK) and 0.2% Normocin cat. #MSPP-ANTNR1 (InvivoGen, Toulouse, France) and transported to the laboratory at approximately 4 °C. Upon arrival in the laboratory, tissue was embedded in low-melting-point agarose, cat. #CSL-LMA (Cleaver Scientific, Warwickshire, UK). Embedded tissue was sectioned at 250 μm using the Compresstome VF-300-OZ system (Precisionary, Natick, MA, USA). In brief, the agarose-embedded tissue was glued onto the plastic holder and carefully placed against the blade in the M199 culture medium bath. The oscillation (setting 6) and the advancement (setting 5) settings of the compresstome were optimised to avoid ‘chatter’, an artefact which manifests as parallel lines of thick and thin sections, and this enabled sectioning of the tissue into uniform 250 μm thick slices. The slices were then placed into a 12-well plate pre-prepared with culturing conditions (DMSO, 10, 20, 30 μg/mL PAN) and put into a 37 °C incubator for 24 h.

The slices were then subsequently fixed in 4% paraformaldehyde (PFA) (*w*/*v*), cat. #28906 (Thermo Scientific™, Rockford, IL, USA) for 1 h at room temperature and stored in PBS, cat. #P4417 (Sigma Aldrich, Paisley, UK) at 4 °C. Treated and fixed tissues were histologically processed and paraffin embedded; additional 2 μm thick sections were prepared for immunofluorescence and imaging.

### 2.3. Cell Culture

A conditionally immortalised LY podocyte cell line was kindly donated by Prof. Moin Saleem (Bristol, UK), whose research group derived it from a 6-year-old patient with a urinary tract infection [21]. The cells were immortalised using the BiCis3 plasmid containing hTERT and SV40 temperature-sensitive transgene. At 33 °C, SV40 is in a wild-type conformation and binds and inactivates p53, allowing growth, whereas at 37 °C, SV40 adopts a mutant conformation, does not bind p53, and therefore leads to growth arrest and differentiation. Proliferating podocytes were cultured at 33 °C, 5% CO_2_ in RPMI supplemented with 10% Fetal Bovine Serum (FBS), 1% Penicillin-Streptomycin (P/S) with the addition of 0.1% 50 μL Insulin-Transferrin-Selenium (ITS), cat. #41400045 (ThermoFisher, Paisley, UK) supplement.

Podocyte numbers were determined by counting using a haemocytometer and seeded at 15,000 cells/cm^2^. Cells were transferred to permissive conditions: 37 °C, 5% CO_2_ in RPMI 1640 Medium, GlutaMAX™, cat. #11554516 (Thermo Fisher, Paisley, UK) supplemented with 2% FBS, 1% P/S and 0.1% ITS (differentiation media). Culture medium was changed every 2–3 days for the 14-day differentiation period.

### 2.4. Protein Extraction and Western Blotting

At the end of the treatment period, cells were lysed with cell lysis buffer, cat. #9803 (Cell Signalling, Leiden, The Netherlands), which included a cocktail of additional protease inhibitors 1× Roche cOmplete™, Mini, EDTA-free, cat. #1183617001 (Merck KGaA, Darmstadt, Germany), 1× PhosSTOP™, cat. #4906845001 (Sigma-Aldrich, Paisley, UK) and AEBSF, cat. #SBR00015 (Sigma-Aldrich, Paisley, UK). Then, 100 μL of complete lysis buffer was added to each 10 cm plate, and cells were scraped off using a sterile cell scraper. The lysates were then collected and transferred to a 1.5 mL Bioruptor^®^ Pico Microtubes tube. The lysates were sonicated using a Bioruptor^®^ Pico bath sonicator (Diagenode, Denville, NJ, USA) using the following settings, 3× (30 s ON, 30 s OFF). The concentration of protein was measured using a BCA protein assay. Lysates were denatured at 95 °C for 5 min. Equal amounts of protein were separated by 10% SDS-PAGE and transferred to polyvinylidene fluoride (PVDF) membranes. Upon transferring the protein to the PVDF membrane, the total protein was stained using the Licor Revert™ 700 Total Protein Stain Kits, cat. #P/N 926-11010 (Licor, Cambridge, UK) as per the manufacturer’s protocol. The membranes were then blocked with Odyssey^®^ Blocking Buffer (TBS) for 1 h and incubated overnight at 4 °C with primary antibodies for NFE2L1, cat #HPA063384 (Atlas Antibodies, Bromma, Sweden) at a concentration of 1:1000, and nephrin, cat #GP-N2 (Progen, Heidelberg, Germany) at a concentration of 1:250. The PVDF membranes were scanned on a Licor Odyssey^®^ CLx using Licor Image Studio™ software (version 1.2) utilising the 700 nm and the 800 nm wavelengths of the imaging system. The exposure time was adjusted to avoid under- and over-exposure, and this was standardised for each antibody used. Licor Empiria Studio^®^ was used to quantify and analyse the expression of proteins. Representative whole-gel blot images are shown in Appendix A.

### 2.5. Cell Fractionation

Following the experimental treatments, cells were washed with cold PBS and maintained on ice. Using the cell fractionation kit, cat. #9038 (Cell Signalling, Cambridge, UK), the cytoplasmic fraction was prepared by suspending the cells in Cytoplasm Isolation Buffer (CIB), followed by vortex for 5 s; this was then centrifuged at 500× *g* for 5 min; subsequently, the supernatant was transferred into a 1.5 mL Eppendorf tube, and this was labelled as a Cytoplasmic fraction. The pellet was suspended using the Membrane Isolation Buffer (MIB) and vortexed for 15 s, followed with a centrifuge for 5 min at 8000× *g*; the supernatant was transferred to a new 1.5 mL Eppendorf tube and labelled as a membranous fraction. The leftover pellet was resuspended in Cytoskeleton/Nucleus Isolation Buffer (CyNIB) and sonicated once using the Bioruptor^®^ Pico sonicator and labelled as the nuclear fraction. These fractions were subsequently prepared for loading into the Western blot using a 3× SDS loading buffer with DTT, as per the protocol described in the kit. 

### 2.6. Multiplex Immunofluorescence

Immunohistochemistry for each individual antibody was performed to confirm the optimal dilution by scrutinizing its specificity, consistency and reproducibility on in-house antibody optimization tissue microarray sections, which have various tumour and normal tissue types. Upon deciding the optimal dilution in IHC, single immunofluorescence was performed using tyramide signal amplification (TSA) and a horseradish peroxidase (HRP) detection system, which is more sensitive than DAB (3,3′-Diaminobenzidine)-HRP detection; representative images of single-IF optimization staining are shown in Appendix A. A heat-induced stripping method was included in between primary antibody visualisation to prevent cross-contamination using BOND Epitope Retrieval Solution 1 (pH 6.0), cat. #AR9961 (Leica Microsystems, Milton Keynes, UK) for 20 min at 95 °C. An individual epitope was tested if any changes occurred due to heat-induced stripping, leading to a finalized 5plex IF protocol. 

For standardisation of 5plex IF, Leica BOND RX autostainer (Leica Microsystems, Milton Keynes, UK) was utilised and an in-house quality control section was included per run. Sections with a thickness of 2 μm of the FFPE (formalin fixed paraffin embedded) were dewaxed, followed by rehydration using absolute alcohol and BOND wash buffer. Endogenous peroxidase activity was quenched by peroxide block solution in BOND polymer refine detection kit, cat. #DS9800 (Leica Microsystems, Milton Keynes, UK). Sections were incubated with serum-free protein block, cat. #X0909302 (Agilent Technologies, Cheshire, UK) in order to remove background noise. The first primary antibody, p57, dilution 1:500, cat. #sc-56341 (Santa Cruz Biotechnology, Heidelberg, Germany), was incubated for 40 min, followed by anti-mouse/-rabbit HRP-conjugated secondary antibody incubation, cat #K400111-2, #K400511-2 respectively (Agilent Technologies, Cheshire, UK) for 40 min. This was visualised using TSA FITC, cat. #NEL741001KT (Akoya Biosciences, Marlborough, MA, USA). Heat-induced stripping was carried out for 20 min at 95 °C using BOND epitope retrieval solution 1 cat#AR9961 (Leica Microsystems, Milton Keynes, UK). These steps were repeated for the second primary antibody NFE2L1, dilution 1:2000, cat. #HPA065424 (Atlas Antibodies, Bromma, Sweden), for disease cohort, Synaptopodin, dilution 1:100 cat. #61094 (Progen, Heidelberg, Germany) for ex vivo cohort and the third primary antibody NFE2L2, dilution 1:100, cat. Ab137550 (Abcam, Cambridge, UK), visualised by TSA cy3, cat. #NEL744001KT (Akoya Biosciences, Marlborough, MA, USA) and TSA cy5 cat. #NEL745001KT (Akoya Biosciences, Marlborough, MA, USA), respectively. After peroxide block and serum-free protein block, the last primary antibody, NQO1, dilution 1:100, cat #HPA007308 (Atlas Antibodies, Bromma, Sweden), was incubated for 40 min, followed by anti-rabbit biotinylated secondary antibody for 40 min, which was visualised by alexa fluor 750 conjugated streptavidin. The sections were counterstained with Hoechst33342 dilution 1:100, cat #H3570 (Thermo Fisher, Paisley, UK) and were mounted in ProLong Gold Antifade mounting medium, cat #P36930 (Thermo Fisher, Paisley, UK).

### 2.7. Direct Immunofluorescence

Podocyte cells were seeded and cultured on glass coverslips in 6-well plates for the duration of the differentiation and treatment periods. Thereafter, coverslips were washed 2 × 5 min in cold PBS and fixed in 4% PFA for 15 min at room temperature; cells were then washed 2 × 5 min in cold PBS and permeabilised with 0.2% Triton X^TM^-100, cat. #X100-5ML (Sigma Aldrich, Paisley, UK) for 5 min on a rocker. The cells were then blocked with 5% Bovine Serum Albumin (BSA), cat. #05470-1G (Sigma Aldrich, Paisley, UK) in PBS-T for 30 min at room temperature. Primary antibodies were diluted in 5% BSA and were incubated overnight at 4 °C on a rocker, followed by a wash in PBS 3 × 5 min on a rocker. Then, secondary antibodies, diluted in 3% BSA and incubated for 60 min at room temperature on a rocker, followed by washing in PBS 5 × 5 min. Cells on coverslips were then mounted on a microscopic slide using Prolong Gold Antifade Reagent with DAPI, cat. #P36931 (Invitrogen, Paisley, UK).

### 2.8. Image Acquisition and Analysis

Immunofluorescence images were captured using Zeiss^®^ Axio Scan. Z1 (Carl Zeiss Microscopy GmbH, Jena, Germany) whole-slide scanner. The scanner was operated using ZEN imaging software version 3.1, utilising the Fluorescence camera Axiocam 712 mono (Carl Zeiss Microscopy, Jena, Germany) with a 3.45 µm pixel size and ZEISS Colibri 7 (Carl Zeiss Microscopy GmbH), a 7-wavelength LED light source. Throughout the experiment, the ZEN software was utilized to optimize and apply custom scanning parameters for a set of multiplexed IF. The captured whole-slide images were saved in the .czi image format. All images were captured with 40× magnification lens with default binning settings. The .czi files were imported into QuPath version 0.2.m9 software [22]. To annotate, analyse and measure various features within the glomerulus, an automated QuPath script was written, using groovy (the scripts are available at https://github.com/MustafaElshani/QuPath_Podocyte (accessed on 20 February 2023)). Upon manual segmentation of glomeruli using the built-in segmentation tools, the “Glomeruli analysis_v2.7” script was run in batch mode on the entire cohort, generating annotations within the glomerulus.

In brief, the first step in the script included segmenting all the nuclei within the glomeruli using the Hoechst channel. Podocyte nuclei were segmented based on p57 positivity, a marker of podocyte terminal differentiation [23]. The immunofluorescence labelling of p57 was carried out in an FITC channel, and above a set threshold, the nuclei were labelled as ‘p57 podocyte’. The script continued to annotate the NQO1-positive areas within the glomeruli as ‘NQO1PosArea’. As NQO1 is expressed in the cell’s cytoplasm and is podocyte specific in the glomeruli [18,24], this was then referred to as the podocyte cytoplasmic area. Upon segmentation of all nuclei and annotation of p57-positive nuclei and the NQO1-positive area, various features were measured within the glomerulus, including NFE2L1 and NFE2L2 in different podocyte cell compartments. The measurements of each of the features were then automatically exported into a database file using the “Export Detection Measurements” script. The positivity threshold for all the channels was determined by staining and scanning of a tissue section without the primary antibody present. This gave autofluorescence values for each channel; therefore, any value above that was considered a positive signal.

### 2.9. Statistical Analysis

All data are presented as mean ± SD, n = 3, unless indicated otherwise. Groups were compared using the Wilcoxon signed-rank test. All analyses were performed using the R package, unless otherwise stated. Correlation was evaluated by Spearman’s correlation coefficient.

## 3. Results

### 3.1. Use of Whole-Slide Image Analysis Allows for Measurements of Protein Expression and Glomeruli Features

Determining protein expression and measurements of features within the glomeruli is a challenge; hence, we set out to develop a strategy to extract these measurements using image analysis on whole-slide scans of renal biopsies. Initially, renal biopsy cases were evaluated by a renal pathologist (D.J.H.) using Haematoxylin and Eosin (H&E); representative images of H&E biopsies are shown in Appendix A. We annotated all identified glomeruli with the presence of podocytes, as determined by the p57 marker. Information on each biopsy can be found in the attached Appendix A. The subsequent section was labelled with a podocyte-specific nuclear marker (p57) [25], NFE2L1, NFE2L2, NQO1, and a nuclei marker (Hoechst 33342) using TSA mIF. Whole-slide immunofluorescence images were scanned and exported to QuPath. The segmentation outlays are illustrated in Figure 1, where Figure 1A shows the whole-slide image with manual annotations of all glomeruli, while Figure 1B–E show the detection of p57, NFE2L2, NFE2L1, and NQO1, respectively, using different fluorescent channels. Figure 1F indicates the automated segmentation of NQO1-positive areas within the glomeruli, and Figure 1G shows the automated segmentation of podocyte nuclei using the p57 antibody. Both the p57 and NQO1 expression patterns are consistent with published data [23,24], confirming them as robust markers of podocyte nuclei and cytoplasm, respectively.

Upon segmentation of all nuclei and annotation of p57-positive nuclei and the NQO1-positive areas, various parameters were measured, as detailed in Table 2. Measurements of each feature were then automatically exported into a database file for further analysis. The positivity threshold for all the channels was determined by staining and scanning a tissue section without primary antibody present. This gave autofluorescence values for each channel; therefore, any value above that was considered a positive signal.

This approach successfully allowed for the simultaneous analysis of podocyte nuclear expression of NFE2L1, NFE2L2 and cytoplasmic NQO1 across the entirety of the cohort, with minimal bias.

### 3.2. NFE2L1 Podocyte Nuclear Protein Expression Is Reduced in Kidney Disease

In order to assess any altered expression of NFE2L1 and NF2EL2 proteins in kidney disease, renal biopsy samples were assessed using mIF and whole-slide image analysis.

NFE2L1 and NQO1 were highly expressed in normal kidney glomeruli, with NF2EL1 expression located predominantly in the podocyte nuclei with diffuse staining in the cytoplasm. NQO1 expression was exclusively cytoplasmic, with staining extending to the podocyte secondary processes. NFE2L2 displayed a diffuse expression pattern within the glomerulus and it was difficult to associate expression with a particular glomerular cell type. p57 staining clearly defined the nuclear podocytes (Figure 2).

The nuclear expression of NFE2L1 in podocytes was found to be reduced by up to 3-fold in biopsies of patients with glomerular disease (Figure 3A). Interestingly, even in cases of minimal-change disease (MCD), which is known to cause podocyte effacement, nuclear NFE2L1 expression was significantly decreased. The mean expression of NFE2L1 was slightly higher in diabetic nephropathy and mesangial IgA cases compared to other renal diseases. A similar pattern was observed with the NQO1 analysis, where NQO1 positivity percentage (Figure 3B) and expression intensity (Figure 3D) were reduced in all disease biopsies. Although changes in NFE2L2 nuclear expression in podocytes between normal and diseased kidneys were not statistically significant, there was a slight upward trend in the distribution in diabetic nephropathy and mesangial IgA groups (Figure 3C).

Since NQO1 is a transcriptional target of both NFE2L1 and NFE2L2 [19], we evaluated whether there was a correlation in expression between either transcription factor and NQO1 intensity. Upon analysing the NFE2L1 and NQO1 expression as a glomerular percentage positivity and its intensity (Figure 4), it was found that these two variables were highly correlated, which is indicative of a relationship between their expression in podocytes.

In contrast, the expression of NFE2L2 and NQO1 showed no significant correlation; Appendix A. Furthermore, we noticed NQO1 aggregation in glomeruli in renal disease biopsies, as illustrated in Appendix A, the significance of which is difficult to evaluate within the scope of this study.

### 3.3. Protein Expression Levels of NFE2L1, Synaptopodin and NQO1 Are Reduced in PAN-Treated Ex Vivo Renal Tissue Slices

To further investigate NFE2L1 and NQO1 expression in podocytes, we used an ex vivo kidney injury model [26]. Slices of normal kidney parenchyma were cultured in DMSO, as a vehicle carrier control, and 20 μg/mL of PAN. After 24 h of treatment, the sections were fixed, histologically processed, and paraffin embedded. The viability of the ex vivo tissue slices was evaluated by a consultant renal pathologist (DJH), representative images of an H&E of untreated and PAN-treated kidney slices are presented in Appendix A. The glomerulus was intact, with attached podocytes visible.

In untreated kidney slices, most glomeruli were intact and expressed synaptopodin covering the entirety of capillary loops. NFE2L1 expression was located in podocyte nuclei; NQO1 expression was higher in untreated slices, compared with treated slices. In PAN 20 μg/mL ex vivo treated slices, most glomeruli had reduced expression of synaptopodin, and no NFE2L1 was observed, even in the presence of p57-positive podocytes. NQO1 intensity was reduced in PAN-treated slices compared with untreated slices, with higher intensity corresponding to synaptopodin expression (Figure 5). Further representative images of ex vivo slices can be seen in Appendix A.

We further evaluated the expression of NQO1 and synaptopodin, and found that synaptopodin did not co-localise with NQO1 (Figure 6). However, since synaptopodin is usually expressed in the podocyte foot processes, these observations suggest that NQO1 expression is mainly located in the cell body.

### 3.4. Protein Expression Levels of Nuclear NFE2L1 Isoform Are Reduced in PAN-Treated Podocyte Cell Line

NFE2L1 is post-translationally modified, giving rise to multiple isoforms, ranging from 95 kDa to 140 kDa. Upon cellular fractionation of differentiated podocytes, it is evident that the fully glycosylated ~140 kDa protein is visible in the membranous/organelle fraction with two bands (Figure 7). Additionally, deglycosylated proteins are visible at ~115 kDa and ~120 kDa, presumably still bound to the ER membrane. A ~95 kDa isoform is observed in the membranous/organelle fraction. In the cytoplasmic fraction, two main bands are visible at ~130 kDa and ~120 kDa. The nuclear fraction is expressed with two nuclear isomers ~95 kDa and ~110 kDa bands. Both isoforms are described as proteolytically derived from full-length NFE2L1α/TCF11 isoforms and are transcriptionally active [27].

We evaluated the protein expression of NFE2L1 in 14-day-differentiated podocytes treated with three concentrations of PAN. Our results show that the expression of 95 kDa isoforms was reduced at all three concentrations, in addition to the 110 kDa isoform, which was barely visible at the 20 and 30 μg/mL PAN concentrations (Figure 8A). Next, we wanted to determine whether treatment with a NFE2L1 inducer, T1-20 [28], would lead to recovery of the expression of the nuclear isoform. We treated podocytes with 20 μg/mL of PAN alone and in combination with T1-20, an NFE2L1 inducer [24]. Our results show increased expression of the 95 kDa isoform in the presence of 0.5 μM of T1-20 (Figure 8B).

Given that nephrin is considered a marker of podocyte health [18], we also evaluated the protein expression of nephrin in podocytes treated with PAN and T1-20. Our results show that exposure to 20 μg/mL of PAN for 24 h led to a reduction in nephrin expression, as expected (Figure 8C). However, treatment of podocytes with a combination of PAN and T1-20 at a concentration of 0.5 μM led to a slight increase in nephrin expression compared to PAN alone.

## 4. Discussion

Podocytes have evolved to reside in challenging environments, where they are terminally differentiated, unable to be replaced, suspended in the urinary space, and constantly being exposed to fluidic pressures and filtered solutes. Harder and colleagues identified approximately ten transcription factors that give podocytes their identity [20], and there is a growing body of evidence suggesting that reduced expression of these transcription factors plays a role in disease [29]. Many of the transcription factors have previously been shown to play a role in podocyte injury and glomerular disease, such as MAFB [30], TCF21 [31], DACH1 [32] and the recently reported FOXC1 [33]. The role of NFE2L1 transcription factor protein expression in podocytes has not previously been reported. In this study, we show that NFE2L1 expression is reduced in kidney disease, and in ex vivo and in vitro models of podocyte injury.

NFE2L1 is a complex multi-isoform transcription factor that undergoes post-transcriptional and post-translational modification, giving rise to differently sized isoforms, each functioning in a feedback loop in multi-process molecular mechanisms [27]. In the context of transcriptional regulation, the NFE2L1 transcription factor is known as the master regulator of the Ubiquitin Proteosome System (UPS). It transcriptionally targets all the UPS isoforms, playing a pivotal role in proteasome homeostasis [34]. In addition, NFE2L1 is thought to be responsible for the basal expression of antioxidant genes [35]. Depending on the cellular context, NFE2L1 has been shown to play a role in cellular differentiation, such as in osteoblasts, where siRNA depletion of NFE2L1 leads to reduced expression of osterix, an essential transcription factor in osteoblast differentiation [36]. This study implicates NFE2L1 in maintaining podocyte health. Firstly, NFE2L1 expression is reduced in podocytes in diseased human kidneys. To further investigate the claim that NFE2L1 is important in podocyte injury, we established an ex vivo kidney injury model. We found that NFE2L1 was expressed in podocytes from untreated normal kidney slices and that when a kidney slice from the same tissue source was exposed to PAN injury, NFE2L1 expression was diminished. NFE2L1 yields complex protein isoforms with different regulator mechanisms; in this study, we characterised the expression profile of these isoforms in differentiated podocytes. We confirmed that two isoforms that are associated with NFE2L1 transcriptional activity were present in the nucleus of podocytes, and that expression of both isoforms was reduced drastically upon PAN-induced injury in podocyte cell lines. We then exposed podocytes to an NFE2L1 inducer, compound T1-20, which attenuated podocyte injury, as indicated by the expression of the podocyte-specific protein nephrin.

Single-cell analysis of podocyte development identified the NFE2L1 transcription factor as a highly expressed gene in mature podocytes [20]. This is further substantiated by single-cell transcriptomics of human foetal kidney cell progenitors [37], where NFE2L1 is expressed in podocytes (Appendix A). In addition, the KPMP Consortium study of the cell states in human kidney [38] showed that mean expression of NFE2L1 RNA was reduced in kidney disease compared to the healthy reference (Appendix A). Considering the reported functions of NFE2L1 in protein homeostasis, antioxidant regulation and its role in inflammation, the podocyte response to injury lends itself to suggesting that NFE2L1 expression must be a vital part of the podocyte’s homeostasis. Widenmaier et al. established that Nfe2l1, while bound to the ER, acts as a sensor for intracellular cholesterol via its CRAC domain in addition to the regulation of CD36, a fatty acid scavenger receptor, expression [39]. Lipid nephrotoxicity and its role in progressive glomerular disease has been reported previously [40], and it is now accepted as a major factor in the development of chronic kidney disease [41]. Cholesterol accumulation is also thought to play a role in podocyte injury [42], mainly due to the increased fatty acid uptake mediated by CD36 and the decrease in ABCA1 expression, thus reducing lipid efflux, hence leading to accumulation of lipids [43]. Furthermore, accumulation of cholesterol has been identified in FSGS [44] and diabetic nephropathy [45]. Therefore, we postulate that lipid accumulation and the effect on NFE2L1 expression in podocytes are contributory factors in the pathogenesis of glomerulopathies; however, further work is required to elucidate the role of cholesterol metabolism and NFE2L1 in the context of podocyte injury.

Additionally, we established that NFE2L1 expression is correlated with that of NQO1 in podocytes of kidney biopsies. NQO1 is highly expressed in normal podocytes, and in our studies, we found that its expression is cytoplasmic; however, not in the podocyte foot processes. NQO1 is usually thought of as an inducible antioxidant gene [46], but unexpectedly, we found no correlation with the NFE2L2 transcription factor. The high expression of NQO1 in podocytes has not been investigated, and little is known regarding its regulation. There is; however, evidence that Nfe2l1 regulates the basal expression of NQO1 in mouse neuronal cells. It has been shown that specific knockout of Nfe2l1 in the central nervous system significantly reduces the basal expression of NQO1 and Glutathione S-transferases (GST) α4 isotype, with a marginal decrease in GSTπ1, which gives rise to severe neurodegeneration. In contrast, specific knockout of Nfe2l2 did not give such a phenotype [47]. Podocytes, in particular, have a very high level of NQO1 expression in a normal basal state; this was reported by Zappa et al. using immunohistochemical staining on normal patient biopsies [18]. In a recent study, Moon et al. found that NQO1 expression played a critical role in maintaining the homeostasis of F-actin in podocytes and that NQO1 deficiency resulted in abnormal synaptopodin localisation [24].

NQO1 is usually reported as a surrogate readout of NFE2L2 transcriptional activity, known as a master regulator of antioxidant genes, and it is mainly thought to be induced upon oxidative stress [46]. Furthermore, it has been reported that podocytes do not express nuclear Nfe2l2 at the basal state. Nfe2l2 knockout mice did not differ in terms of nephrin expression from wild-type mice [48]; however, upon treatment with nephrotoxic agent, Nrf2 knockout mice presented with a more severe glomerular injury and that genetic activation of Nfe2l2 by Keap1 knockout attenuated glomerulosclerosis. It can be postulated that Nfe2l2 activation is a response to certain conditions and acts as a cytoprotective; however, this does not explain the high basal expression of NQO1.

In the context of glomerular disease, there have only been a handful of studies published regarding NFE2L2 stress response pathways in podocytes. High glucose is used as an in vitro experimental model mimicking diabetic conditions, and it is utilised as a stress injury model in podocytes due to the high generation of reactive oxygen species in response to it [49]. It has been shown that the addition of the potent NFE2L2 activator tert-Butylhydroquinone (tBHQ) was able to attenuate hyperglycaemic mediated injury in immortalised mouse podocyte cell lines. This was evidenced by the nuclear accumulation of Nfe2l2 and increased expression of antioxidant proteins, including NQO1. The podocyte-specific protein, synaptopodin, also increased, recovering to normal control levels after treatment with tBHQ in high glucose. It was not, however, possible to maintain the recovery of podocytes under the high glucose conditions due to the deleterious effects of glucose on the activity and function of antioxidant proteins [50]. Furthermore, pharmacological activation of Nfe2l2 in the streptozotocin (STZ) diabetic mouse model has been shown to attenuate the characteristics of diabetic nephropathy, including reduced albuminuria, renal hypertrophy and thickening of the basement membrane [51].

A study that included podocyte-specific genetic and pharmacological inhibition of GSK3β lead to Nfe2l2 nuclear accumulation and a potent antioxidant response after doxorubicin injury. This significantly ameliorates signs of podocyte injury, including loss of podocyte-specific markers and ultrastructural changes such as foot process effacement [52]. Additionally, another study that looked at Nfe2l2 pathway activation using podocyte-specific knockdown (KD) of Keap1 in mice showed that injury alone did not activate Nfe2l2. The reason for these somewhat conflicting results can be explained by GSK3β overexpression in podocyte injury [53] and its subsequent prevention of Nfe2l2 nuclear translocation. It should be noted, however, that a recent clinical trial of a potent NFE2L2 inducer, bardoxolone methyl, known as the BEAM study, showed some improvements in eGFR of patients with type 2 diabetes with Chronic Kidney Disease (CKD) in the initial phase of the trial, but did not show any improvement in Phase III and had to be abandoned due to cardiovascular complications [54]. The unsuccessful results of this trial with bardoxolone could be due to the effects it has on other targets apart from Nrf2 and the complex array of pathways that Nrf2 regulates. Accumulating evidence strongly supports that the expression of Nfe2l2 transcriptional targets such as NQO1, GSTs, HO-1 and SOD have a cytoprotective role in podocytes. However, their expression initiated by NFE2L2 transcription factor is thought to be in response to oxidative stress rather than maintaining podocyte homeostasis. 

In this study, we characterized the expression of multiple NFE2L1 protein isoforms in podocytes and demonstrated that NFE2L1 nuclear expression is reduced in podocytes from kidney disease patient biopsies and in injured podocyte cell lines. Further work is needed to investigate the intricate molecular regulation of NFE2L1 and its proteoforms, as well as to identify downstream transcriptional targets in podocyte cells.

## Figures and Tables

**Figure 1 cells-12-02165-f001:**
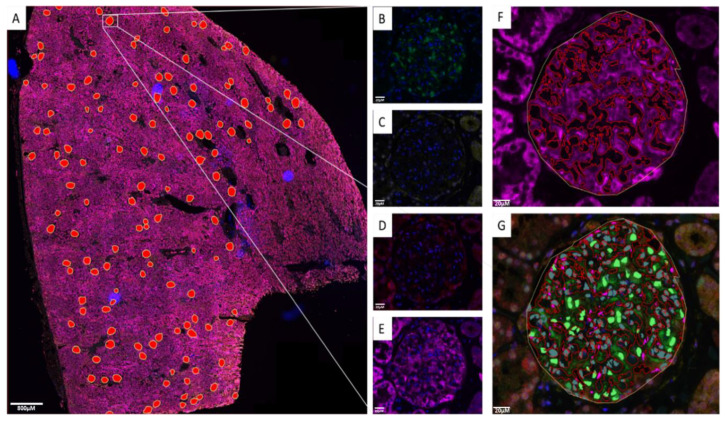
QuPath image analysis. (**A**) Whole-slide image of 5-plexed nuclei (Hoechst 33342), p57(FITC), NFE2L2(Cy3) NFE2L1(Cy5) and NQO1(AF750) showing manual annotation of glomeruli shown as red overlays. (**B**) p57 detected in the FITC channel showing podocyte-specific nuclear labelling. (**C**) NFE2L2 detected in the Cy3 channel. (**D**) NFE2L1 detected in the Cy5 channel. (**E**) NQO1 detected in the AF750 channel. (**F**) Automated labelling of NQO1-positive area within the glomeruli indicating the podocyte cell body (**G**) Automated podocyte nuclear segmentation using podocyte-specific p57 antibody. Scale bar for (**A**) 800 μm; scale bar for (**B**–**G**) 20 μm.

**Figure 2 cells-12-02165-f002:**
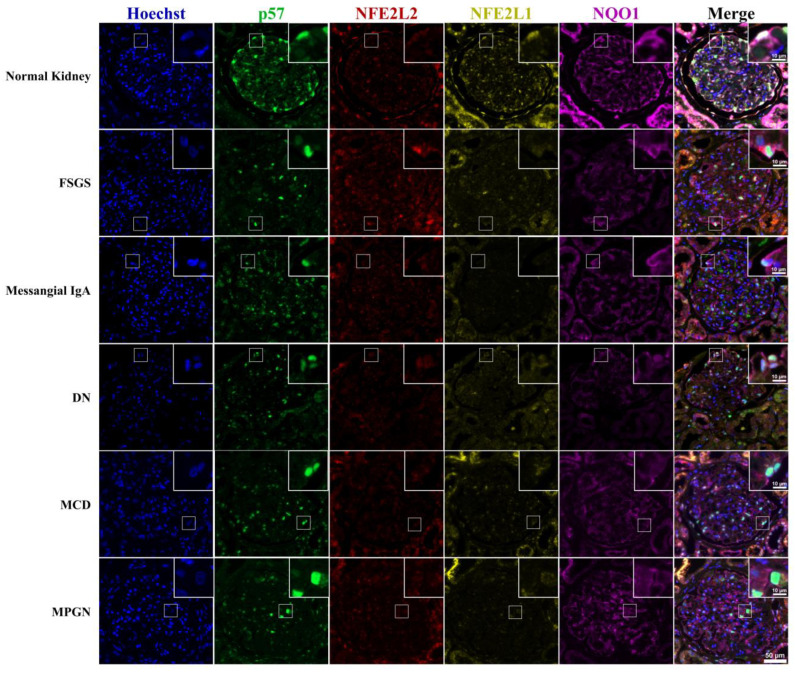
NFE2L1, NFE2L2, p57 and NQO1 immunofluorescence staining of renal disease biopsy. Representative images of multiplexed staining of renal biopsy cases. The top right corner inset image is a magnified image showing a podocyte cell, Hoechst (blue) nuclei marker, p57 (green) podocyte marker, NFE2L2 (red), NFE2L1 (yellow), NQO1 (purple). Inset scale bar = 10 μm; image pane scale bar = 50 μm.

**Figure 3 cells-12-02165-f003:**
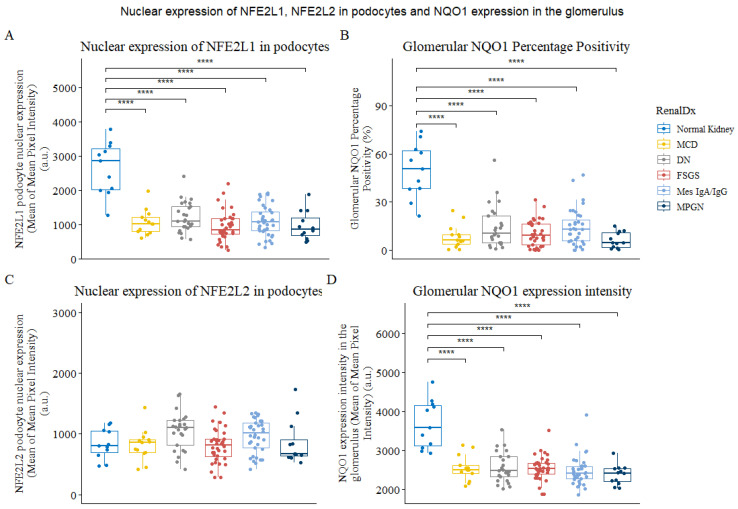
Expression of NFE2L1, NFE2L2 and NQO1 in podocytes of normal and diseased kidneys. (**A**) Plot for mean of mean NFE2L1 expression in podocyte nucleus in whole-slide images from normal kidney parenchyma and renal disease biopsies. (**B**) Glomerular NQO1 percentage positivity. (**C**) Plots for mean of meanNFE2L2 expression in podocyte nucleus from normal kidney parenchyma and renal disease (**D**) The plot shows NQO1 expression intensity in the glomerulus. Each data point represents a biopsy or normal tissue parenchyma from renal cancer nephrectomies categorised as normal kidney. Comparison was performed using Wilcoxon’s signed-rank test. **** represents *p* < 0.0001.

**Figure 4 cells-12-02165-f004:**
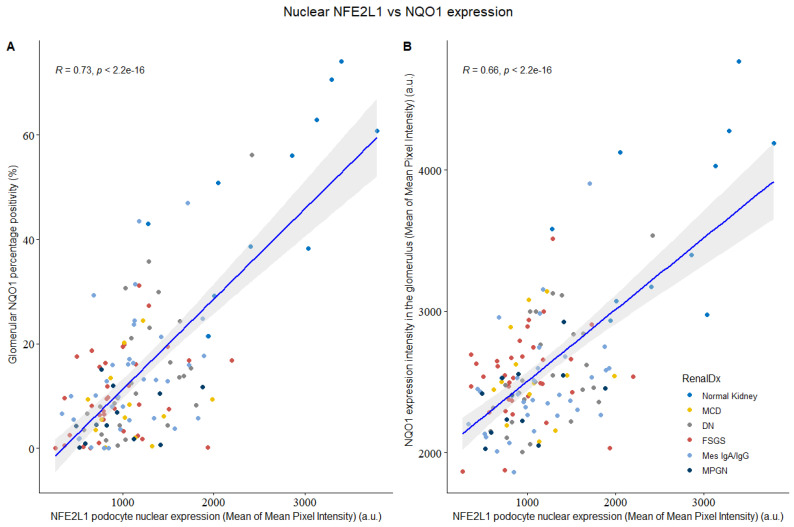
NFE2L1 podocyte nuclear expression vs. glomerular NQO1 intensity from normal kidney and disease biopsies. (**A**) Comparison between NFE2L1 nuclear expression and glomerular NQO1 percentage positivity shows signification correlation (R = 0.73, *p* < 2.2 × 10^−16^). (**B**) Comparison between NFE2L1 podocyte nuclear expression and NQO1 expression intensity shows signification correlation (R = 0.66, *p* < 2.2 × 10^−16^). Correlation was evaluated by Spearman’s correlation coefficient.

**Figure 5 cells-12-02165-f005:**
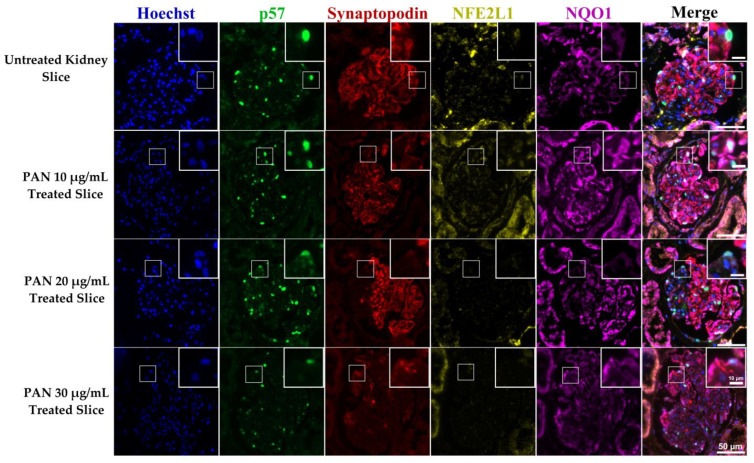
NFE2L1, NQO1 and synaptopodin expression in untreated and PAN-treated ex vivo human kidney slices. A representative image of a glomerulus from ex vivo kidney slices, untreated slices from normal kidney parenchyma and treated with concentrations of 10, 20 and 30 μg/mL for 24 h. Sections were multiplexed with Hoechst (blue), p57 (green), synaptopodin (red) NFE2L1 (yellow) and NQO1 (purple) antibodies. Magnification = 20× with scale bar representing 20 μm.

**Figure 6 cells-12-02165-f006:**
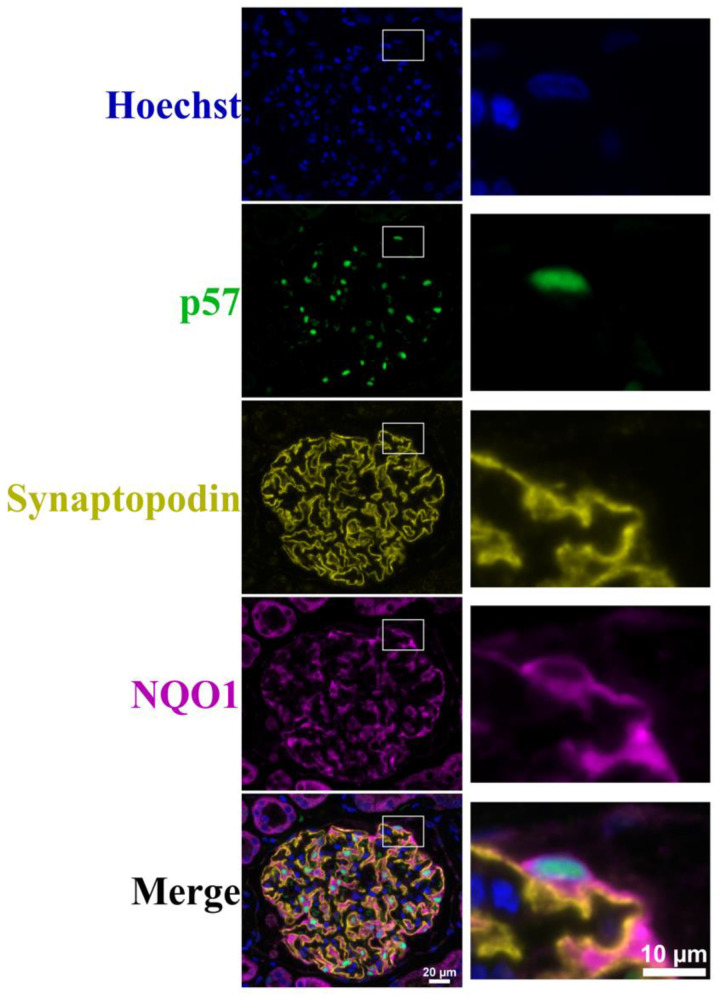
Hoechst, p57, synaptopodin and NQO1 immunofluorescence staining of ex vivo tissue. A representative image of glomeruli from untreated ex vivo kidney slice multiplexed with Hoechst (blue) nuclear staining, podocyte nuclear marker p57 (green), synaptopodin (yellow) and NQO1 (purple). Scale bar represents 20 μm. Images on the right represent zoomed-in images of the marked white box. Scale bar represents 10 μm.

**Figure 7 cells-12-02165-f007:**
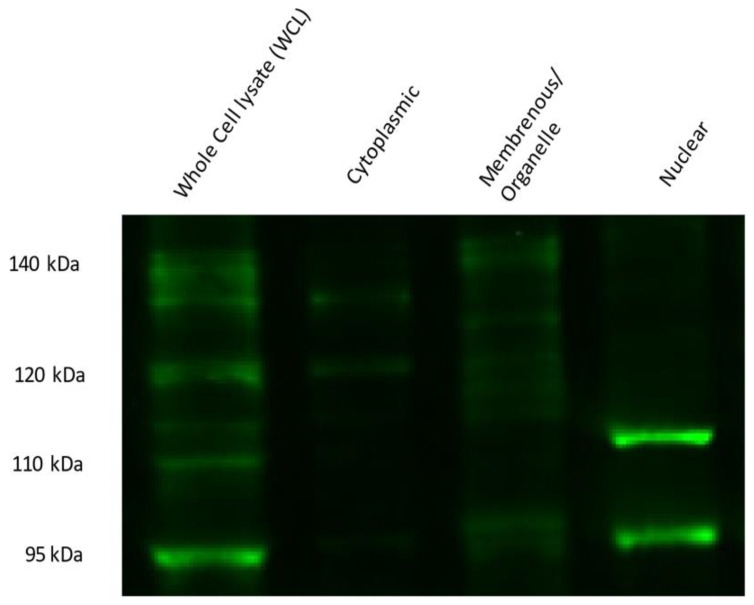
NFE2L1 protein isoform expression in differentiated podocyte cellular fractions. Briefly, 10% SDS PAGE of 14-day-differentiated podocytes cell fractions were probed with NFE2L1 antibody; 40 μg of protein was loaded for whole-cell lysates, while 20 μg of lysate was loaded for each fraction.

**Figure 8 cells-12-02165-f008:**
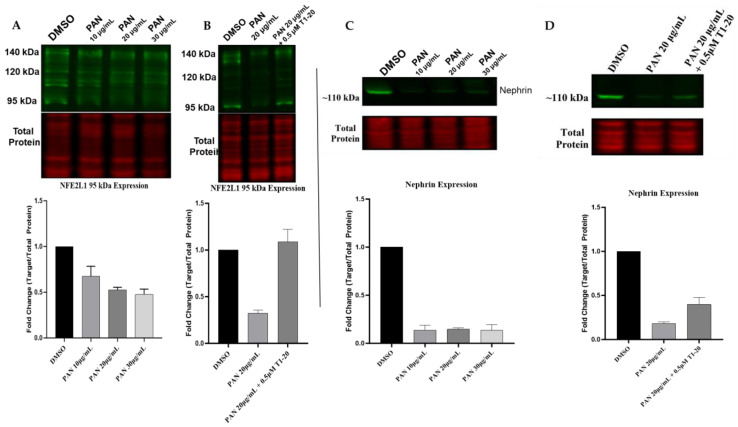
NFE2L1 nuclear isoform is reduced in PAN-treated podocytes. (**A**) Representative Western blot image of the NFE2L1 protein and the total protein of 14-day-differentiated podocytes treated with PAN at concentrations of 10 μg/mL, 20 μg/mL and 30 μg/mL for 24 h. (**B**) The blot shows NFE2L1 protein expression in 14-day-differentiated podocyte samples of untreated control, DMSO and samples treated with 20 μg/mL PAN for 24 h ± T1-20. (**C**) Representative Western blot image of nephrin protein and the total protein of 14-day-differentiated podocytes treated with concentrations of 10 μg/mL, 20 μg/mL and 30 μg/mL PAN for 24 h. (**D**) Representative image of nephrin protein expression in 14-day-differentiated podocyte samples of untreated control, DMSO and samples treated with 20 μg/mL PAN for 24 h ± T1-20. All protein expression levels are relative to the total protein staining for the same blot. Fold change quantification is a representation of 3 biologically independent experiments for the 95 kDa isoform.

**Table 1 cells-12-02165-t001:** Kidney disease cohort characteristics.

Characteristic		All (n = 127)	MCD	DN	FSGS	MesIgA/IgG	MCGN
Sex	F	52 (40.9%)	3 (23.1%)	7 (25.9%)	19 (51.4%)	18 (48.6%)	5 (38.5%)
	M	75 (59%)	10 (76.9%)	20 (74.1%)	18 (48.6%)	19 (51.4%)	8 (61.5%)
* Age		49.0 ± 15.8	51.2 ± 21.2	62.2 ± 11.8	52.0 ± 17.5	44 ± 15.4	49.3 ± 13.6
* s-Creatinine (µmol/L)		229.5 ± 192.7	110.9 ± 113.1	203.7 ± 115.4	183.2 ± 105.2	184.5 ± 216.0	225.8 ± 156.6
* Urea (mmol/L)		12.2 ± 7.6	9.1 ± 8.7	13.1 ± 6.7	10.7 ± 5	10.3 ± 9.3	12.1 ± 4.6
* Albumin		27.6 ± 18.5	16.7 ± 16.4	24.6 ± 16.9	24.9 ± 16.6	31.1 ± 18.6	27.3 ± 19.2
* eGFR (mL/min/1.73 m^2^)		35.5 ± 8.3	53.3 ± 7.6	34.7 ± 10.9	36.7 ± 10.6	44.4 ± 7.0	33.3 ± 3.5
* Blood Glucose (mmol/L)		7.3 ± 4.3	7.3 ± 3.0	11.3 ± 6	6.1 ± 1.6	5.8 ± 3.6	5.33 ± 0.6

* Values are mean ± SD.

**Table 2 cells-12-02165-t002:** Description of QuPath measured features.

Feature	Description
Number of glomeruli	Sum of annotated glomeruli per case
Area of annotated glomeruli	Mean area of all annotated glomeruli (μm^2^)
Number of podocytes	Sum of p57-positive podocytes per case
Podocytes per glomerulus	Total number of p57-positive podocyte/total number of glomeruli in a case
Area of NQO1 positivity.	Mean of NQO1-positive area within glomerulus (μm^2^)
Glomerular NQO1 percentage positivity	Mean percentage of NQO1-positive area/total glomerulus area (%)
NQO1 expression intensity	Mean of total mean NQO1 pixel intensity within NQO1-positive area in the glomerulus
NFE2L2 intensity within NQO1-positive area	Mean of total mean NFE2L2 pixel intensity within NQO1-positive area in the glomerulus
NFE2L1 intensity within NQO1-positive area	Mean of total mean NFE2L1 pixel intensity within NQO1-positive area in the glomerulus
Podocyte nuclear area	Mean of Podocyte nuclear area (μm^2^) area in a case
p57 podocyte nuclear expression	Mean of total mean p57 pixel intensity within podocyte nucleus
NFE2L1 podocyte nuclear expression	Mean of total mean NFE2L1 pixel intensity within podocyte nucleus
NFE2L2 podocyte nuclear expression	Mean of total mean NFE2L2 pixel intensity within podocyte nucleus

## Data Availability

All data needed to evaluate the conclusions in the paper are present in the paper and/or the Appendix A. Additional data are available from the authors upon request.

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
