# Peer review of "Transcription Factor NFE2L1 Decreases in Glomerulonephropathies after Podocyte Damage"

_cells, 2023, doi:10.3390/cells12172165_

Round 1

Reviewer 1 Report

In this manuscript by Elshani, et al, entitled “Transcription factor NFE2L1 maintains podocyte differentiation” the authors employ elegant multiplex imaging combined with automatized analysis workflows to assess a large cohort of well-characterized kidney biopsies. The authors focus on the transcription factors NFE2L1 (NRF1), NRF2 as well as NQO1 (as a downstream target) and aim to elucidate the expression and localization patterns in glomerular disease. NFE2L1 showed here (independent of the disease) reduced expression levels in podocytes. Moreover, the authors employ acute ex vivo slice culture techniques to demonstrate that acute podocyte injury (PAN application) is related to reduced expression levels. Finally, the authors demonstrate in a podocyte cell line that active isoforms of NFE2L1 are present in the nucleus in vitro and also respond to PAN-damage. 

In summary, the authors provide a very comprehensive and detailed description of NFE2L1, NRF2 and NQO1 expression levels in a large cohort of well-characterized kidney biopsies. These studies are complemented by some additional lines of evidence showing that podocyte damage is related to reduced expression levels of NFE2L1.

The descriptive part of the study (including multiplex imaging and establishing the quantitative analysis workflows) is innovative and (given the large cohort) an impressive piece of work. On the other hand, functional analysis related to diminished levels of NFE2L1 are somewhat missing (but might be beyond the scope of this study). However, the authors should clearly reduce their claims in terms of functional relevance of this reduced expression levels (including the title of this study - this would require further genetic evidence including knockdown or knockout studies). 

Specific points:

Figure 2: 

-No images regarding the entities MCD and/or mesangioproliferative GN are provided.  

-H&E, PAS stains should be provided (at least in the supplemental section) in order to correlate IF patterns with structural alterations (in particular FSGS). 

-FSGS measurements appear somewhat difficult as these lesions tend to be very focal and segmental. The authors should comment how many glomeruli per biopsy have been assessed and how the analysis was performed. 

-NQO1 expression has been reported to be increased in models of DN (STZ) – (e.g. Moon, et al, Scientific Reports 2020 – also cited by the authors). The authors should comment on these conflicting results (used antibodies?).

-Using this level of magnification/resolution definite localization patterns (e.g. secondary podocyte processes as mentioned by the authors) cannot really be assessed – this should be corrected/adapted in the manuscript.

-How about localization of NFE2L1/NRF2/NQO1 in other glomerular cell types? Any changes here in the investigated glomerular diseases?

-Did the authors correlate the findings with available transcriptomic data (NephroSeq) or other available scRNASeq data (in terms of transcriptional alterations?).

Figure 3: 

-The figure legend appears not to be correct. There is a description of image data, which are not presented in the actual figure

-How are the scores (e.g. nuclear expression score) generated? How is the unit derived (µm/pixel)?

Figure 4: 

-The authors mention an accumulation of NQO1. Given the actual image data this is difficult to evaluate. The authors should provide more representative imaging data and should also discuss their observations (e.g. in the discussion section)

Figure 5&6: 

-The figure panel should be rearranged in order to demonstrate the dose response reaction of the kidney tissue (moreover histology H&E or PAS staining of the tissue should be provided to asses the impact of culture conditions as well as treatment - the exemplary data in the supplemental section could/should be extended, eg. overviews, more glomeruli).

-Moreover, it would be nice to see the established analysis workflow for quantitative data (see figures before) applied on the tissue slices (quantitative data would be very valuable).

Figure 8:

-Western blot of treated podocytes -> the 95 kDA target band is almost not to visible on the presented cropped images (uncropped western blot data should be provided in the supplemental section)

-Are there statistically significant differences in the densitometric western blot data?

-Are there also changes detectable on the mRNA level (would be interesting to see for NFE2L1 and NEPHRIN)?

-Very interesting observation that the NRF-inducer ameliorates the reduced expression level of NFE2L1 – how about using this inducer on the tissue slices to demonstrate the effect in this ex vivo setup?

-Are there any changes detectable for NRF2?

Minor:

-       2.2. incomplete sentence/typo (second line “ .).”)

-       Quantitative data on the measurements should be provided -> e.g. number of glomeruli analyzed per sample (ideally in table form).

Reviewer 2 Report

In the present study by Elshani et al., the authors investigated expression of Nuclear Factor Erythroid 2-related Factor 1 (NFE2L1), a transcription factor, and NAD(P)H, quinone oxidoreductase (NQO1) in podocytes. They performed multiplex immunostaining of renal biopsy specimen from glomerular disease patients and quantified the expressions by image analysis. They found that both NFE2L1 and NQO1 were significantly reduced in the glomeruli of disease biopsies. They also showed reduction of NFE2L1 in kidney slices or podocyte in culture exposed to Puromycin Aminonucleoside (PAN). From these observations the authors concluded that reduction of NFE2L1 is a common finding observed in vivo, ex vivo, and in vitro associated with glomerular disorders. However, the main data are not firm enough to support the conclusion.

Therefore, the present study would not be acceptable for publication in Cells.

Concerns are as follows.

Major comments

1) For quantification, various measurements were performed using QuPath, and the measurements were defined as “Feature” and explained as “Description” in table 2.

For example, three types of measurements were performed for NQO1. They appear in table 2 as follows.

・     meanNQO1Glom: Mean of NQO1 positive area within glomerulus (μm2)

・     meanNQO1GlomAreaPrc: Mean percentage of NQO1 positive area/total glomerulus area (%)

・     meanOfmeanNQO1intensity : Mean of total mean NQO1 pixel intensity within NQO1 positive area in the glomerulus.

However, these definitions are hardly used in the text, figures, and figure legends. Instead, for NQO1 measurements, the following annotations are used. But these expressions are not defined nor explained.  

・     Glomerular NQO1 percentage positivity (Figure 3B)

・     glomerular percentage positivity of NQO1(Figure 3B Figure legend)

・     Glomerular NQO1 expression intensity (Figure 3D)

・     mean expression intensity of NQO1 (Figure 3D, Figure legend)

・     NQO1 positive Glomerular Area (%) (Figure 4A)

・     NQO1 percentage positivity within the glomerulus (Figure 4A, Figure legend)

・     Glomerular NQO1 intensity (Figure 4B )

・     NQO1 intensity (Figure 4B, Figure legend)

Because of these inconsistent expressions and annotations, it is impossible to understand the analyses in the results accurately.

2) line 222: The segmented nuclei are then used to detect the positivity in the p57 FITC channel, and above a set threshold, the nuclei are labelled as ‘p57 podocyte’. The script continues to annotate the NQO1 positive areas within the glomeruli as ‘NQO1PosArea’. As NQO1 is expressed in the cell’s cytoplasm and is podocyte-specific, this was then referred to as the podocyte cytoplasmic area.

As described in Materials and Methods, authors identified podocyte nuclei and podocyte cytoplasmic area by immunoreactivity for p57 and NQO1 respectively. First of all, they should justify these criteria. Percentage of podocyte nuclei in total nuclei in each glomeruli would be constant in normal kidney. Podocyte cytoplasmic area would be the same. These figures should be shown and confirmed to be compatible with those in previous reports. Next, it should be clarified how the number and size (=cytoplasmic area) of podocyte are changed in each glomerular disease. In Figure 2, number of podocyte nuclei (p57 positive dots), and podocyte cytoplasmic area (NQO1 positive area) seem drastically reduced. But these reductions are not quantitatively analyzed. These analyzes should be shown before comparing protein expressions quantitatively (Figure 3).

3) Figure 2, Figure3, Figure S2

Immunofluorescence for Nrf2 are extremely weak even in normal kidney (Figure 2). Based on this immunostaining, authors show that Nrf2 expression in disease is not changed (Figure 3C). In addition, they describes as follows “In contrast, the expression of Nrf2 and NQO1 showed no significant correlation, Figure S1” (line 320). It is supposed that the immunofluorescence data for Nrf2 was included as a positive control, which expression is not affected by glomerular dysfunction. However, authors should consider whether Nrf2 is an appropriate positive control as it’s expression is low. If it is, immunostaining condition for Nrf2 should be optimized.

4) The titleTranscription factor NFE2L1 maintains podocyte differentiation” seems inappropriate. Even if NFE2L1 expression is specifically decreased in podocyte in the glomerular diseases, the finding does not necessarily means that NFE2L1 plays a role to maintain podocyte differentiation.

5) Figurer 1A should be shown without manual annotation of glomeruli. As it is, it is impossible to know how much the proteins immunostained are concentrated at the glomeruli. Also, negative control for the immunostaining should be shown here or as a supplementary figure to prove the specificity of the immunostaining.

Minor points

1) Figure S3

The area enclosed with a rectangle in A does not match with the enlarged image shown in B.

2) Figures 3, 4, 8, S1, S2, S4

Letterings in the figures are too small to read, even on the computer screen.

3) Line 647

Reference 35: The title is written in capital letters.

Reviewer 3 Report

In this manuscript the Authors studied the expression pattern of NFE2L1 and NQO1 in podocyte both in human biopsies than in podocytes culture. They demonstrated a decreased expression in nephropathies. This downregulation appears to be independent from the pathogenesis of nephropathy. To this purpose different glomerulonephritis (MCD, DN, FSGS, MCGN) were analyzed.

The main actor appears to be the podocyte. The knowledge of new pathogenetic mechanisms and consequently the involvement of different molecules appears to be very interesting as it could help in the approach of new therapeutic targets.

Reviewer 4 Report

This manuscript describes the protein expression of the transcription factors NFE2L1 (NRF1) and NFE2L2 (NRF2) in human kidney disease and specifically in podocyte cells. Podocyte health is critical for maintaining the filtration barrier and thus kidney function. They also study the expression of NQO1 as it is a transcriptional target of NFE2L1 and NFE2L2. The authors used immunofluorescence on human renal biopsies from different diseases (minimal change disease, diabetic nephropathy, focal and segmental glomerulosclerosis, among others), podocyte cell culture, ex-vivo renal tissue slices and western blot methods. In summary, they conclude from their study that NFE2L1 protein expression is significantly decreased in renal diseases compared to controls, while NFE2L2 protein expression was not much altered. They also study that NFE2L1 expression can be prevented during injury with the use of an NFE2L1 inducer and thus that NFE2L1 is essential I podocytes.

This study is interesting and well described, but I am however not sure I am fully convinced about the findings.

Major concerns:

-          What was the goal of doing all the first part, generating the Table 2, for not using the data ? I was expecting to see some correlation of quantitative measurements with protein expression. In particular the number of podocytes. It is well known that there is podocyte loss in kidney diseases, so maybe the decrease of protein expression that is observed is only a reflection of the podocyte loss. Is there a way to “normalize” Figure 3 to the number of podocytes ?

-          Page 9 mentions accumulation of NQO1; if the transcription factor NFE2L1 decreases, should we not expect less expression of NQO1 ? Without quantification of Figure 5, I feel difficult to make the statement that NQO1 expression is decreased in PAN treated slides (which seems contradictory with the page 9 statement) (also on page 10 is repeated the information: “NQO1 expression was higher in untreated slides” / “NQO1 intensity was reduced in PAN treated slides”.

Minor concerns:

-          The reader must wait page 9 to understand why the authors looked at NQO1 (transcriptional target of both NFE2L1/NFE2L2). Please provide this information at the beginning of the story.

-          It would be nice to be consistent throughout the paper by using either the terms NFE2L1/NFE2L2 or NRF1/NRF2 (instead of NFE2L1/NRF2 or NRF1/NFE2L2).

-          In the result subtitles, I would mention that the authors are looking at protein expression (e.g. 3.3. Protein expression levels of NFE2L1).

-          Figure 4: The figure legend mentions “from normal and disease biopsies”: it would be nice to graphically be able to identify the normal vs disease in general (e.g. 2 colors). Please increase the size of the Figure 4 graph axis legend.

-          Even though it is known that regulatory processes happen between mRNA and protein expression, I would suggest to the authors to compare their results with recent data publicly available such as the KPMP single nuc (podocytes well represented)/single cell/spatial transcriptomics data.

-          Figure 3 legend: ”mean of the mean” ?????

-          Figure 8: legend of C and D are missing. It is weird to quickly mention Nephrin only at the end. There is a typo in the word “Nephrin” on the right of Figure 8C.

Round 2

Reviewer 1 Report

The authors addressed all raised questions. 

Author Response

Thank you for taking the time to review our revised manuscript and for acknowledging that we addressed all the raised questions. We appreciate your valuable feedback throughout this process, which has significantly improved the manuscript

Reviewer 2 Report

The revised version of the study by Elshani et al. was largely improved and some concerns were cleared. However, the following concern (“Major comments 2 in the previous review”) is not cleared. This point is the most essential comment I made in the last review, and I strongly encourage the authors to clear this.

The original comment, the author’s response (yellow highlight), and my response (green highlight)

(original comment)

2) line 222: The segmented nuclei are then used to detect the positivity in the p57 FITC channel, and above a set threshold, the nuclei are labelled as ‘p57 podocyte’. The script continues to annotate the NQO1 positive areas within the glomeruli as ‘NQO1PosArea’. As NQO1 is expressed in the cell’s cytoplasm and is podocyte-specific, this was then referred to as the podocyte cytoplasmic area.

As described in Materials and Methods, authors identified podocyte nuclei and podocyte cytoplasmic area by immunoreactivity for p57 and NQO1 respectively. First of all, they should justify these criteria. Percentage of podocyte nuclei in total nuclei in each glomeruli would be constant in normal kidney. Podocyte cytoplasmic area would be the same. These figures should be shown and confirmed to be compatible with those in previous reports. Next, it should be clarified how the number and size (=cytoplasmic area) of podocyte are changed in each glomerular disease. In Figure 2, number of podocyte nuclei (p57 positive dots), and podocyte cytoplasmic area (NQO1 positive area) seem drastically reduced. But these reductions are not quantitatively analyzed. These analyzes should be shown before comparing protein expressions quantitatively (Figure 3).

(author’s response)

You raise a valid point regarding the constant percentage of podocyte nuclei in total nuclei and the consistency of podocyte cytoplasmic area in normal kidneys. We agree that these figures should have been shown and confirmed to be compatible with previous reports. However, in this particular study, our primary focus was on investigating the nuclear expression of NFE2L1 in podocytes and its relation with NQO1. As such, we did not extensively analyze other glomerular cell types.

Additionally, we understand the importance of assessing changes in the number and size of podocytes in different glomerular diseases. Unfortunately, disease needle biopsy proved challenging, and obtaining sufficient samples to perform detailed analyses of podocyte numbers and cytoplasmic areas across various glomerular diseases was beyond the scope of this study.

Nevertheless, we wholeheartedly agree with your observation that the number of podocyte nuclei (p57 positive dots) and podocyte cytoplasmic area (NQO1 positive area) appeared reduced in Figure 2. Given the constraints of this study, we are grateful for your understanding and acknowledgment of the primary aim and focus of our research. We aimed to investigate the nuclear expression of NFE2L1 in podocytes and its relationship with NQO1, as stated in the study objectives. Moving forward, we will carefully consider your feedback and suggestions when planning future investigations or follow-up studies.

(new comment)

As the author replayed, investigating the expression of NFE2L1 in podocytes and its relation with NQO1 is the main object of this study. However, the expression is of NFE2L1 is quantified using p57 and NQO1 as standard indexes of podocyte nuclei and podocyte cytosol, respectively. It is not required to examine whether expressions of p57 and NQO1 are altered in various glomerular diseases. But, it would be required to demonstrate that p57 and NQO1 are present in podocyte nuclei and podocyte cytosol, respectively, in normal kidney. If this has been reported already, just citing appropriate previous studies would suffice. If not, double immunostaining of normal kidney with podocyte markers, such as synaptopodin or podocin, would work. Alternatively, podocyte cell lines, such as MPC (Mouse Podocyte) HPC (Human podocyte), could be immunostained for p57 and NQO1. In this way, the following description at Line 275 “As NQO1 is expressed in the cell’s cytoplasm and is podocyte-specific, this was then referred to as the podocyte cytoplasmic area.” would be justified.

Other minor comments

1) Authors unified a protein name changing “Nrfr2” to “NFE2L2” in the text. But “NRF2” is left in the abstract.

2) In Table 2. “mm2” should be “mm2”.

3) Some features do not have units. (e.g. “NQO1 expression intensity” in Figure 3 A, C, and D). For these graphs, arbitrary unit (AU, a.u.) could be used.

Author Response

Comments and Suggestions for Authors

The revised version of the study by Elshani et al. was largely improved and some concerns were cleared. However, the following concern (“Major comments 2 in the previous review”) is not cleared. This point is the most essential comment I made in the last review, and I strongly encourage the authors to clear this.

The original comment, the author’s response (yellow highlight), and my response (green highlight)

(original comment)

2) line 222: The segmented nuclei are then used to detect the positivity in the p57 FITC channel, and above a set threshold, the nuclei are labelled as ‘p57 podocyte’. The script continues to annotate the NQO1 positive areas within the glomeruli as ‘NQO1PosArea’. As NQO1 is expressed in the cell’s cytoplasm and is podocyte-specific, this was then referred to as the podocyte cytoplasmic area.

As described in Materials and Methods, authors identified podocyte nuclei and podocyte cytoplasmic area by immunoreactivity for p57 and NQO1 respectively. First of all, they should justify these criteria. Percentage of podocyte nuclei in total nuclei in each glomeruli would be constant in normal kidney. Podocyte cytoplasmic area would be the same. These figures should be shown and confirmed to be compatible with those in previous reports. Next, it should be clarified how the number and size (=cytoplasmic area) of podocyte are changed in each glomerular disease. In Figure 2, number of podocyte nuclei (p57 positive dots), and podocyte cytoplasmic area (NQO1 positive area) seem drastically reduced. But these reductions are not quantitatively analyzed. These analyzes should be shown before comparing protein expressions quantitatively (Figure 3).

(author’s response)

You raise a valid point regarding the constant percentage of podocyte nuclei in total nuclei and the consistency of podocyte cytoplasmic area in normal kidneys. We agree that these figures should have been shown and confirmed to be compatible with previous reports. However, in this particular study, our primary focus was on investigating the nuclear expression of NFE2L1 in podocytes and its relation with NQO1. As such, we did not extensively analyze other glomerular cell types.

Additionally, we understand the importance of assessing changes in the number and size of podocytes in different glomerular diseases. Unfortunately, disease needle biopsy proved challenging, and obtaining sufficient samples to perform detailed analyses of podocyte numbers and cytoplasmic areas across various glomerular diseases was beyond the scope of this study.

Nevertheless, we wholeheartedly agree with your observation that the number of podocyte nuclei (p57 positive dots) and podocyte cytoplasmic area (NQO1 positive area) appeared reduced in Figure 2. Given the constraints of this study, we are grateful for your understanding and acknowledgment of the primary aim and focus of our research. We aimed to investigate the nuclear expression of NFE2L1 in podocytes and its relationship with NQO1, as stated in the study objectives. Moving forward, we will carefully consider your feedback and suggestions when planning future investigations or follow-up studies.

(new comment)

As the author replayed, investigating the expression of NFE2L1 in podocytes and its relation with NQO1 is the main object of this study. However, the expression is of NFE2L1 is quantified using p57 and NQO1 as standard indexes of podocyte nuclei and podocyte cytosol, respectively. It is not required to examine whether expressions of p57 and NQO1 are altered in various glomerular diseases. But, it would be required to demonstrate that p57 and NQO1 are present in podocyte nuclei and podocyte cytosol, respectively, in normal kidney. If this has been reported already, just citing appropriate previous studies would suffice. If not, double immunostaining of normal kidney with podocyte markers, such as synaptopodin or podocin, would work. Alternatively, podocyte cell lines, such as MPC (Mouse Podocyte) HPC (Human podocyte), could be immunostained for p57 and NQO1. In this way, the following description at Line 275 “As NQO1 is expressed in the cell’s cytoplasm and is podocyte-specific, this was then referred to as the podocyte cytoplasmic area.” would be justified.

Thank you for your reply and clarification, we confirm that p57 and NQO1 are well known markers nuclei and cytoplasm, respectively. Therefore we have added the appropriate reference where suggested.

Other minor comments

1) Authors unified a protein name changing “Nrfr2” to “NFE2L2” in the text. But “NRF2” is left in the abstract.

 Thank you, this has now been rectified in the manuscript.

2) In Table 2. “mm2” should be “mm2”.

Thank you, this has now been rectified.

3) Some features do not have units. (e.g. “NQO1 expression intensity” in Figure 3 A, C, and D). For these graphs, arbitrary unit (AU, a.u.) could be used.

Thank you, this has now been rectified in the manuscript.

Reviewer 4 Report

No more comments.

Author Response

(The authors gave the same response as above.)

Round 3

Reviewer 2 Report

I found that the authors added new references on p57 and NQO1. Since localizations of p57 and NQO1 have been well characterized in these papers, the authors should briefly describe in Results or Discussions whether localizations for p57 and NQO1 in the present study are compatible or inconsistent. The response of this part in the rebuttal letter, as below,does not tell how authors modified the text and referred the papers.   "Thank you for your reply and clarification, we confirm that p57 and NQO1 are well known markers nuclei and cytoplasm, respectively. Therefore we have added the appropriate reference where suggested."   Other points are OK
